# Identification of Recurrent Chromosome Breaks Underlying Structural Rearrangements in Mammary Cancer Cell Lines

**DOI:** 10.3390/genes13071228

**Published:** 2022-07-11

**Authors:** Natalie C. Senter, Andrew McCulley, Vladimir A. Kuznetsov, Wenyi Feng

**Affiliations:** 1Department of Biochemistry and Molecular Biology, SUNY Upstate Medical University, 750 East Adams Street, Syracuse, NY 13210, USA; nwagaman1@gmail.com (N.C.S.); a.mcculley12@gmail.com (A.M.); kuznetsv@upstate.edu (V.A.K.); 2Department of Urology, SUNY Upstate Medical University, 750 East Adams Street, Syracuse, NY 13210, USA

**Keywords:** breast cancer, spontaneous chromosome breaks, double-strand breaks (DSBs), CNV, 16q loss, structural rearrangements, genome instability, MCF-7, MCF-10A, Break-seq, SHCBP1, ORC6

## Abstract

Cancer genomes are characterized by the accumulation of small-scale somatic mutations as well as large-scale chromosomal deletions, amplifications, and complex structural rearrangements. This characteristic is at least partially dependent on the ability of cancer cells to undergo recurrent chromosome breakage. In order to address the extent to which chromosomal structural rearrangement breakpoints correlate with recurrent DNA double-strand breaks (DSBs), we simultaneously mapped chromosome structural variation breakpoints (using whole-genome DNA-seq) and spontaneous DSB formation (using Break-seq) in the estrogen receptor (ER)-positive breast cancer cell line MCF-7 and a non-cancer control breast epithelium cell line MCF-10A. We identified concurrent DSBs and structural variation breakpoints almost exclusively in the pericentromeric region of chromosome 16q in MCF-7 cells. We fine-tuned the identification of copy number variation breakpoints on 16q. In addition, we detected recurrent DSBs that occurred in both MCF-7 and MCF-10A. We propose a model for DSB-driven chromosome rearrangements that lead to the translocation of 16q, likely with 10q, and the eventual 16q loss that does not involve the pericentromere of 16q. We present evidence from RNA-seq data that select genes, including SHCBP1, ORC6, and MYLK3, which are immediately downstream from the 16q pericentromere, show heightened expression in MCF-7 cell line compared to the control. Data published by The Cancer Genome Atlas show that all three genes have increased expression in breast tumor samples. We found that SHCBP1 and ORC6 are both strong poor prognosis and treatment outcome markers in the ER-positive breast cancer cohort. We suggest that these genes are potential oncogenes for breast cancer progression. The search for tumor suppressor loss that accompanies the 16q loss ought to be augmented by the identification of potential oncogenes that gained expression during chromosomal rearrangements.

## 1. Introduction

Chromosomal alterations such as deletions, amplifications, fusions, and translocations are a hallmark for many cancer genomes, particularly breast and colorectal cancers [1,2,3]. The identification of recurrent breakpoints of these structural alterations is important for understanding the mechanism of carcinogenesis, discovering prognostic and diagnostic biomarkers, and for revealing drug targets. Currently, genome-wide detection of chromosome fusion or translocation junctions largely derives from whole-genome DNA sequencing or microarray-based comparative genome hybridization. However, these junctions do not necessarily correspond to the initial chromosome breakage site that give rise to subsequent chromosome translocants. One of the reasons for this discrepancy is that an initial DNA double-strand break (DSB) might undergo extensive processing by the DNA repair enzymes before fusion or translocation occurs. Moreover, many such junctions identified in the DNA sequencing data might prove to be false-positive events without validation by an independent method. Thus, it stands to reason that identifying recurrent chromosome breakage sites using DNA DSB mapping technologies will allow us to understand what gives rise to chromosome translocations. We posit that simultaneous mapping of DSBs and chromosome translocation breakpoints is a necessary approach that provides an additional and essential layer of complexity to the understanding of cancer genome evolution.

Here we employed a technology named Break-seq, with demonstrated efficacy in human lymphoblastoid cell lines [4,5], to map DNA DSBs using massive parallel sequencing. We coupled this DSB mapping with whole-genome DNA sequencing (DNA-seq) to systematically compare spontaneous, but recurrent, DSBs with chromosome structural variant breakpoints. We further complemented this combined approach with RNA-seq to measure potential gene expression changes near the chromosome breakage sites. Such a combinatorial genome-wide comparison of static structural variation breakpoints with recurrent DSBs has not been performed before. We believe this approach allows one to test if the structural breakpoints represent recurrent DSBs that are still contributing to genome rearrangements, or are mere by-products of past rearrangement events. As a proof-of-principle we selected a malignant estrogen receptor (ER)-positive breast cancer epithelial cell line, MCF-7, and a non-malignant breast epithelial control cell line, MCF-10A, for comparison. MCF-10A and MCF-7 are representative cell lines of a lineage model of differentiation that tracks the epithelial cell hierarchy, from a basal-like mammary stem cell line (MCF-10A) to a luminal B breast cancer cell line (MCF-7) [6]. Moreover, MCF-7 is arguably the most widely used breast cancer cell line, and its genome was characterized by CGH, SKY, and End-sequencing of BAC libraries to ascertain structural breakpoints [7,8,9]. These previous studies provide the basis for some key characteristics of the MCF-7 cell line, making it a highly suitable model for the current study.

Early studies detected specific chromosome structural changes that stratify breast cancer genomes with varying degrees of gross anomalies [10]. Among these, rearrangements involving chromosomes 1 and 16, which predominantly led to 1q gain and 16q loss, were frequently associated with tumors with few gross anomalies, suggesting that 1q gain and 16q loss are early chromosomal alterations in breast cancer [8,10,11,12,13,14,15]. Subsequently, 16q loss has been shown to be associated with low genetic grade, luminal subtype breast cancer and better prognosis [16], suggesting that the dynamic alterations of 16q are correlated with cancer development as well as progression. However, the mechanism with which 16q loss is induced is still unclear. Specifically, the precise location of the breakpoint on 16q that leads to its loss has not been described. Therefore, we set out to determine the recurrent DSBs in the MCF-7 cancer genome using Break-seq.

We identified 297 recurrent DSBs specifically in the MCF-7 cancer genome. We further identified structural breakpoints, including DNA deletions, amplifications, and translocations, in the MCF-7 genome through DNA-seq analysis, and the associations of gene expression data with these chromosome alterations. The overlaps between the recurrent breaks and the structural breakpoints in MCF-7 are exclusively located in the pericentromeric region on chromosome 16q. Our results, therefore, suggest chromosome breakage as an underlying mechanism of gross chromosome alterations in the cancer genome.

## 2. Materials and Methods

### 2.1. Cell Lines and Growth Conditions 

MCF-7 cell line was obtained from the Sheikh laboratory at Upstate Medical University, which obtained the cell line from NIH. MCF-10A cell line was purchased from ATCC. MCF-7 cells were grown in DMEM (Corning#10-013-CV, Corning, NY, USA) media containing 10% FBS (GeminiBio#100-106, West Sacramento, CA, USA) and 1X Penstrep (Corning#30-002-CI, Corning, NY, USA). Cells were split at 70–80% confluency and expanded to obtain >4 × 10^7^ cells before harvest, wherein 3 × 10^7^ cells were used for Break-seq, RNA-seq, and DNA-seq for two technical replicates (see below), and the remaining 10^7^ cells were used for further passage for a biological replicate experiment. MCF-10A cells were grown in medium171 (Life Technologies#W171500, Carlsbad, CA, USA) containing MEGS (Life Technologies #S0155, Carlsbad, CA, USA). Cells were split at 95–100% confluency and expanded to obtain >4 × 10^7^ cells as described above for MCF-7 cells. Samples were collected from two biological replicates each, by two technical replicates.

### 2.2. Break-Seq 

Two replicates of 10^7^ cells were harvested, and 5 × 10^6^ cells were used to make each agarose plug for Break-seq experiments. Detailed Break-seq procedures were previously described [5]. Briefly, 5 × 10^6^ cells were embedded into 0.5% Incert low-melting point agarose in PBS and cast into plug mold. The agarose plugs were then incubated at 50 °C overnight in 6 mL of lysis buffer (0.5 M EDTA, 1% Sarkosyl, 200 μg/mL Proteinase K). The DNA in the agarose plugs was then end-labelled in-gel using the End-It enzyme (Epicentre Biotechnologies, Madison, WI, USA) with biotinylated dNTP mix (1 mM dTTP, dCTP, dGTP, 0.84 mM dATP, 0.16 mM Biotin-14-dATP). Plugs were then treated with β-Agarase (NEB, Ipswich, MA, USA) to digest agarose and release DNA. The DNA sample was then sonicated using a Covaris M220 using the snap-cap DNA 300 bp shearing protocol. DNA was then processed using a PCR Clean-up Kit (Qiagen, Germantown, MD, USA) and run on agarose gel to verify the fragmentation pattern of DNA, and quantified on a Nanodrop. Following this, 10–11 μg of DNA was end repaired and purified using the PCR Cleanup Kit. The DNA was then A-tailed by A-tail Kit (NEB, Ipswich, MA, USA) or Klenow exo- (NEB#E6054A, Ipswich, MA, USA) and purified using the PCR Clean-up Kit, followed by quantification on a Nanodrop. M270 Dynabeads (Life Technologies, Carlsbad, CA, USA) were used to purify biotinylated DNA. The amount of DNA bound to beads was calculated by measuring the quantity of DNA in the flow-through. DNA-bound beads were then resuspended in ligation mix containing Illumina adaptors (50 μM adaptor-1, 50 μM adaptor-2, 1X T4 ligase buffer, 3 μL T4 DNA ligase) and incubated overnight at room temperature on a roller. Following this, 400 ng of DNA bound to beads was used for PCR amplification using KAPA Hotstart Ready Mix (Roche, Branchburg, NJ, USA). Each sample was given a specific index primer for multiplexing. PCR product was then run on agarose gel to verify amplification and quantity. AMPure beads (Agencourt Bioscience, Beverly, MA, USA) were used to remove free adaptors, and the final product was analyzed on agarose gel. Break-seq libraries were sequenced on Illumina Hi-Seq 2500.

### 2.3. DNA-Seq 

Two replicates of 10^7^ cells were harvested, and 5 × 10^6^ cells were used to make each agarose plug for DNA-seq. Genomic DNA was isolated from agarose plug to simulate the exact conditions for Break-seq samples, and to provide an appropriate control. DNA isolation was performed identically to Break-seq samples, except that the first step of End-repair with biotinylated dNTPs was omitted. After sonication, 10 µg DNA was End-repaired in a 100-μL reaction (1x End-repair buffer (33 mM Tris-acetate, pH 7.8, 66 mM potassium acetate, 10 mM magnesium acetate, and 0.5 mM dithiothreitol), 250 μM dNTP mix, 1 mM ATP, and 3 μL End-It enzyme (Epicentre Biotechnologies, Madison, WI, USA) for 45 m at room temperature. DNA was purified using a Qiagen PCR purification column and eluted in 32 μL elution buffer. The End-repaired DNA was then A-tailed using standard conditions with Klenow (NEB E6055A, Ipswich, MA, USA) and Qiagen column purified. A-tailed DNA was used to ligate adaptors with T4 DNA ligase (ligation reaction mix containing 30 nM each of Solexa Adapter 1 and Adapter 2, incubated at room temperature overnight with agitation) and purified using a Qiagen purification column in 50 μL elution buffer. DNA was then amplified by PCR (18 cycles) using PlexP1 primer and Illumina Index primers (2, 5, 6, and 12) (98 °C, 5 m, followed by 18 cycles of 98 °C, 20 s; 65 °C, 15 s; 72 °C, 1 m; followed by 72 °C, 5 m). The PCR products were purified using AmPure beads (Agencourt Bioscience, Beverly, MA, USA) to remove free adapters according to the manufacturer’s recommendations; the resulting DNA was subjected to paired-end (2 × 150 bp) sequencing on the Illumina Nextseq-500 platform. Copy numbers were determined using the R package “QDNAseq” using default parameters [17]. Structural breakpoints prediction was performed with Socrates using default parameters [18]. Visual presentations of structural alterations obtained using Socrates were compiled on a Circos Table Viewer (http://mkweb.bcgsc.ca/tableviewer/, accessed on 1 July 2019) using default parameters. Specifically, paired translocation events, both intra- and inter-chromosomal, were first output from Socrates. The number of events was tallied for each chromosome and displayed on a Circos plot.

### 2.4. RNA-Seq

Two replicates of 10^7^ cells were harvested, and 5 × 10^6^ cells were harvested for RNA-seq. Total RNA was extracted using the Qiagen RNeasy Plus Mini Kit. The RNA was run on an Agilent 2100 Bioanalyzer using the RNA 6000 Nano Chip to assess RNA quality and quantity. Following this, 1 µg of total RNA was used as input for the Illumina TruSeq Stranded Total RNA Library Prep Kit Ribo Zero Gold H/M/R. Library size was assessed using the DNA 1000 chip on the Bioanalyzer, and the libraries were quantified using a Qubit fluorometer. Pair-end sequencing was run on an Illumina NextSeq 500 instrument. Four replicates were processed for treatment/conditions, out of which three were biological replicates. Raw reads were obtained from Illumina; Base space and pair-end reads were merged. Merged sequence reads were then aligned to the UCSC human genome assembly, GRCh37/hg19, using STARfusion aligner. The BAM files generated by STARfusion were then subjected to featureCounts [19] for generation of read counts per gene. The RNA-seq expression count obtained from featureCounts was Log2 transformed. Significance was determined using one-way ANOVA. The Benjamini–Hochberg method was used to calculate the false discovery rate (FDR). Significant differentially expressed genes (DEGs) were determined at a *p*-value <= 0.05 using edgeR (Empirical Analysis of Digital Gene Expression Data in R). Up-regulated and down-regulated genes were determined by a fold change (MCF-7 relative to MCF-10A) of >1 and <1, respectively.

### 2.5. Survival Prediction Analysis

To identify the survival significance of experimentally defined variables (as a function of average gene expression level in the cell samples), we carried out a survival prediction analysis. This analysis is based on the concept of a data-driven grouping [20,21] that proposes using survival data and experiment outcomes to estimate the cut-off value(s) of the prognostic variables (e.g., gene expression level) that allow splitting the patients cohort into two (or more) risk groups. In this study, we identified the candidate genes using the survival data and Affymetrix microarray expression profiles of 4934 breast cancer patients available in the Kaplan–Meier (K–M) plotter (https://kmplot.com/analysis/index.php?p=service&cancer=breast, accessed on 1 May 2022). We also used K–M plotter tools for survival prediction analysis. Breast cancer patients were stratified into relatively low- and high-risk groups via optimization of cut-off value defined computationally over gene expression signal data in the tumor samples of the studied cohort. Recurrence-free survival (RFS) time was used as the endpoint of the prognosis model outcomes.

### 2.6. Gene Ontology

Gene ontology analyses were performed using DAVID Bioinformatics tools (https://david.abcc.ncifcrf.gov, accessed on 4 July 2019).

## 3. Results

We profiled a malignant breast epithelial cell line, MCF-7, with a non-malignant breast cell line MCF-10A as control. We applied Break-seq to map spontaneous DNA DSBs in cells grown in normal culture conditions (see Methods for details). We performed four independent replicate Break-seq experiments per cell line, as well as two replicate RNA-seq experiments, and one DNA-seq experiment, per cell line. The overall experimental design and analysis pipeline is shown in Figure 1. The central questions we wished to address were: (1) Where are the cancer-specific spontaneous chromosome breakage hotspots? (2) Do these cancer-specific chromosome breakage hotspots correspond to CNV or structural breakpoints?, and (3) Do these breaks correlate with cancer-specific gene expression changes? By asking these questions, we hoped to identify genetic loci that show concurrent DSB formation, structural breakpoint, and gene expression changes. These loci will potentially provide cancer-specific genetic signatures, further permitting the discovery of tumor oncogenes or suppressors.

### 3.1. High Level of Gene-Associated Spontaneous Chromosome Breakage in the MCF-7 Cell Line

We obtained, on average, 30 million paired-end reads from each of the four replicate Break-seq libraries from the MCF-7 cell line, and an average of 20 million reads per Break-seq library for MCF-10A. DSB peak locations were obtained via peak calling using MACS2, after copy number normalization using DNA-seq data (see Methods for details). We consistently detected a greater number of spontaneous DSBs in the MCF-7 cells than in MCF-10A cells from all four replicate experiments (Figure 2A), with an average of 2875 DSBs in the MCF-7 cells and 2232 DSBs in the MCF-10A cells. We then extracted consensus DSBs that were identified in every replicate experiment for each cell line, 472 for MCF-7 and 271 for MCF-10A (Figure 2B). Between these consensus DSBs there were 172 common breaks, and 297 and 68 breaks specific to the MCF-7 and MCF-10A cells, respectively (Appendix A). Some of these MCF-7-specific DSBs are discrete sites, as in the *BCAR1* and *CKM* loci (Figure 2C); others are apparently clustered, and span a large genomic region, as in the *DOK5* locus on chromosome 20 (Figure 2C). The vast majority (>90%) of these DSBs map to intergenic regions and introns, followed by promoters, regions immediately downstream of genes, and exons (Figure 2D). No DSB was found in the 5′- or 3′-UTR regions. Only one DSB in MCF-7 cells, and none in MCF-10A cells, was found in the exons. This was a DSB peak spanning base pairs 24,087,493 to 24,087,908 on chromosome X (chrX), which overlaps with exon 9 of *EIF2S3*, encoding the eukaryotic initiation factor 2 subunit 3. Compared to MCF-10A cells, there was a disproportionate increase in DSBs in the genic regions, and exclusively in introns, in the MCF-7 cells (20 and 142 intronic DSBs in MCF-10A and MCF-7, respectively; *p* = 1.8 × 10^−^^14^ in Fisher’s Exact test). This result suggested that genic regions were at higher risk for spontaneous breakage in the MCF-7 cancer cell line.

### 3.2. Concurrent Cancer-Specific Spontaneous DSBs and Structural Variation Breakpoints on the Pericentromere of 16q

We then focused on the MCF-7-specific DSBs and examined their chromosomal distribution. Chromosome 16 had the highest density of DSBs (2.32 per Mb of DNA), whereas chromosome 13 had the lowest coverage of DSBs (0.17 per Mb of DNA) (Figure 2D). The chromosomal view showed extensive DNA breakage throughout chr16 in MCF-7 cells compared to the MCF-10A control cells (Figure 3A). DSB hot spots clustered near the pericentromeric regions, particularly 16q, in both cell lines (purple solid triangles in Figure 3B). In the same pericentromere region on 16q there was an additional cluster of DSBs found only in the MCF-7 cells (orange open triangles in Figure 3B). Another common DSB was found approximately 2.5 Mb downstream from the pericentromeric cluster, and the remainder of the 16q arm exclusively contained DSBs only found in MCF-7 cells (Figure 3B). These results suggested that 16q underwent recurrent breakage near the centromere in both cell lines, but the vast majority of the 16q arm only experienced breakage in MCF-7 cells.

This prediction was directly tested via whole genome sequencing, which we performed to serve a dual purpose: (1) identifying copy number variation and structural variation breakpoints; and (2) providing a normalization control for Break-seq analysis. We obtained paired-end sequences with 3.5× and 3.7× genome coverage for MCF-10A and MCF-7 cell lines, respectively. We detected extensive chromosome copy number variations in MCF-7 cells, including the deletion of 16q (Figure 3B and Figure 4). Interestingly, sequence loss was confined to approximately the centromeric-proximal half of the 16q arm. The immediate pericentromeric region of approximately 300 kb DNA, as well as the telomere-proximal half of 16q, were intact when compared to the MCF-10A control (Figure 3B). This result supported the notion that the structural changes on 16q in MCF-7 cells were the result of recurrent spontaneous breakage. It also suggested that the DSBs in non-cancerous cells did not lead to apparent copy number variation. We then asked if any of the DSBs bore consequences on the structural variation of 16q. First, we detected 1016 and 1374 paired (both strands of the breakpoint were identified) structural variants in MCF-10A and MCF-7 cells, respectively. These variants, after excluding those involving Y chromosomes, mitochondria sequences and unassigned contigs, were displayed on Circos plots (Figure 5A,B). This result suggested that genomic rearrangements were prevalent even in the MCF-10A genome; however, the MCF-7 genome exhibited an ~30% increase in structural variation. We then focused on those variants detected in the MCF-7 cells: 1190 intra-chromosomal and 136 inter-chromosomal (translocation) variants (Appendix A). Chromosomes 1, 3, 5, 11, and 17 were among the top chromosomes with an increased number of rearrangements in MCF-7, while chr16 displayed a similar number of structural variants between MCF-10A and MCF-7 (Figure 5B). Therefore, we then asked where in the genome structural variations overlap with DSBs.

Remarkably, the overlaps between the structural variation breakpoints and the DSBs in MCF-7 cells were almost exclusively located in the pericentromeric region of 16q (four out of five intra-chromosomal junctions and five out of six inter-chromosomal junctions). Additionally, one DSB on chr14, spanning 88,889,718 and 88,890,124, was involved in intra-chromosomal translocation, and one DSB on chr11:51,590,333–51,591,355 overlapped with two breakpoints on chr11 that translocated to chr7:132,676,151 and chr8:135,456,020. These results suggest that spontaneous chromosome breakage only accounts for a small fraction of structural changes that lead to chromosomal translocations in the MCF-7 cancer cell line. Nevertheless, the near exclusive overlap between the cancer-specific DSBs and chromosome translocation breakpoints in the pericentromere of 16q underscore the importance of this region in breast cancer development. Therefore, we next asked if there are gene expression changes in this region in the MCF-7 cells.

### 3.3. Genes Immediately Downstream from the Pericentromere of 16q Showed High Expression in MCF-7 Cells

We performed RNA-seq experiments to examine the transcriptome of the MCF-7 cell line compared to the MCF-10A control. Relative changes in gene expression level were expressed as the Log_2_ value of the fold change (FC), i.e., the ratio of MCF-7 level to that of MCF-10A (Appendix A). The DSB clusters in the pericentromere of 16q did not actually overlap with any genes. Genes located downstream from the DSBs include *SHCBP1*, *VPS35* (head-to-head anti-sense gene overlapping *ORC6*), *ORC6*, and *MYLK3*. All genes except *VPS35* showed significantly higher expression in the MCF-7 cells (Figure 3B, gene with Log_2_FC values >3.5 or <−3.5 are shown). Interestingly, both *SHCBP1* and *ORC6* also showed elevated expression in breast tumor samples compared to control samples in TCGA studies (Appendix A, FC = 7.42 and 5.48, respectively). This pattern of increased gene expression no longer persisted downstream and ended at *MYLK3*. We wondered if this was the product of complex chromosome changes on 16q. The translocation events on 16q presumably involved the swapping of 16q or a portion thereof with 7q (at chr7:61,794,599) and 10q (at chr10:42,393,859 and chr10:43,034,287). All three breakpoints on chr7 and chr10 were also located in the pericentromeric regions. We speculated the complex structure of 16q post-translocation and proposed a model in which the translocants such as t(10;16) now contain euchromatic pericentromeric sequences, thus promoting the over-expression of genes downstream (Figure 3C).

### 3.4. SHCBP1 and ORC6 Are Effective Predictive and Poor Prognosis Markers for (ER)-Positive Breast Cancer Patients

To clarify the clinical significance of *SHCBP1* and *ORC6*, we carried out a survival prediction analysis of these genes using Affymetrix microarray expression profiles in 4934 breast cancer patients available in the Kaplan–Meier (K–M) plotter. Recurrence-free survival time was used as the endpoint of prognosis. The patients were stratified into relatively low- and high-risk groups via optimization of the cut-off value defined computationally over gene expression signal data in a studied group. We hypothesized that the observed structural variation breakpoints in MCF7 cells, and observed over-expression of *SHCBP1* and *ORC6,* affect the outcomes of breast cancer patients whose primary tumors are ER-positive. K–M plots (Figure 6A,B) showed that *SHCBP1* and *ORC6* genes are strong, clinically relevant prognostic factors in the ER-positive patient cohort, but not in the ER-negative cohort (Figure 6C,D). We also observed that over-expression of *SHCBP1* and *ORC6* mRNAs is significantly associated with poor outcomes, suggesting pro-oncogenic properties of both genes (Figure 6A,B). We then investigated the potential clinical roles of *SHCBP1* and *ORC6* mRNAs in the contexts of different treatment approaches. We restricted the ER-positive patient groups with (i) endocrine therapy + adjuvant only and (ii) endocrine therapy + neoadjuvant only. Figure 6E,H show a comparison of K–M survival functions that separate the patients into relatively low- and high-risk groups with high confidence levels. We found that the expressions of the genes in different breast cancer groups and tumor subtypes are positively correlated (r > 0.5, *p* < 0.0001, Spearman correlation coefficient). Furthermore, our results demonstrate that *SHCBP1* and *ORC6* can serve as prediction markers for the essential impact on disease progression by treatment, both providing a larger value of Hazards Ratio (HR) and increasing PFS time, specifically in low-risk groups of ER-positive breast cancer patients (Figure 6E,H).

### 3.5. Genes Associated with MCF-7-Specific DSBs Were Enriched in Biological Pathways including the ER Signaling Pathway

We also asked if genes located near recurrent chromosome breaks show changes in expression in the MCF-7 cells compared to the MCF-10A control. We identified differentially expressed genes (Log_2_FC values > 4 or <−4) in the two cell lines. Under these criteria, there were 1130 up-regulated genes and 1902 down-regulated genes in MCF-7 cells. In parallel, we identified the nearest genes to the 297 recurrent chromosome breaks occurring specifically in MCF-7 cells. Genes within 5 kb distance from the MCF-7-specific DSBs were enriched for those in the “intracellular estrogen receptor signaling pathway” (fold enrichment 23.2, *p* = 0.082). We found two genes (*DOK5* and *ADAMTS10*) that are up-regulated in MCF-7 cells (compared to MCF-10A) and four genes (*KCNQ5*, *KLHL41*, *CAPZA3*, and *PID1*) that are down-regulated, located near a DSB. *DOK5* and *PID1* are both involved in the insulin response pathway. *PID1* was recently identified as one of the two signature genes (the other gene is *SPTBN2*) in seven cancers [22]. *ADAMTS10* is a member of the disintegrin-like and metalloprotease with thrombospondin type I motif family proteins that are implicated in breast cancer development and progression [23]. Together with *ADAMTS3*, *ADAMTS10* expression appeared reduced in breast cancer tissues [24].

## 4. Discussion

We performed the first simultaneous mapping and comparative analysis of high-resolution structural breakpoints and recurrent DSBs in two strategically chosen cell lines of mammary origin. This study was designed to specifically test the hypothesis that genomic regions harboring recurrent DSBs are responsible for structural rearrangements that lead to pathological alterations in the cancer genome. By comparing the ER-positive MCF-7 cancer cells to the MCF-10A control cell line we identified cancer-specific DSBs, or chromosome breakage hotspots. Overall, there are more DSBs in the cancer cell line than the control; under a stringent criterion we defined 472 and 271 consensus DSBs in the MCF-7 and MCF-10A cell lines, respectively. More than 90% of DSBs are located in regulatory regions, including intergenic and intronic regions. However, MCF-7 cells showed a disproportionately higher level of genic DSBs than the control, suggesting a potential transcription-driven mechanism for, and a more detrimental consequence of, DSB formation in the cancer genome. Consistent with this idea, we found that genes associated with the cancer cell-specific DSBs were enriched in the ER signaling pathways. This result suggests that recurrent DSBs may underlie the transition from normal to luminal B cancer cell in the epithelial cell lineage, and ultimately, the pathophysiology of breast cancer progression.

We also investigated the relationship between DSBs and CNV breakpoints to test the hypothesis that recurrent DSBs lead to structural changes in the chromosome. We first verified that MCF-7 (but not MCF-10A) cells underwent 16q loss; however, the sequence loss was confined to approximately half of the 16q arm that is centromere-proximal. We further identified those genomic regions with overlapping MCF-7-specific DSBs and CNV breakpoints. Remarkably, 9 of 11 such locations were located in the pericentromeric region of 16q. In contrast, while MCF-10A cells also sustained DSB formation, it apparently did not lead to 16q loss. Inspecting the gene expression levels by RNA-seq, we detected a small cluster of genes immediately downstream of the pericentromere of 16q showing significant overexpression in the MCF-7 cells compared to control. This pattern of gene expression led us to propose complex chromosome exchange events on 16q which involve swapping 16q sequences with those of 7q and 10q. We surmised that such an exchange may result in the euchromatinization of the 16q pericentromere and enhanced gene expression. Importantly, two of the over-expressed genes, *SHCBP1* and *ORC6*, in this cluster also show significantly increased expression in breast cancer tissues reported in TCGA. We propose that *SHCBP1* and *ORC6* are novel oncogenes involved in breast cancer development. Shcbp1 is an Src homolog and collagen homolog (Shc) SH2-domain binding protein; its overproduction was shown to promote tumor (including breast) cell proliferation, migration, and invasion through multiple signaling pathways such as the cyclin-dependent kinase signaling pathway and the TGF-ß1/Smad signaling pathway [25,26,27,28,29,30,31,32,33,34,35,36,37]. Notably, it was shown that *SHCBP1* was significantly up-regulated in breast cancer tissues, and that *SHCBP1* knockout inhibited cell proliferation [37], thus supporting the hypothesis that *SHCBP1* is an oncogene. Recently, it was shown that the overproduction of HER2-SHCBP1-PLK1 diminishes the efficacy of trastuzumab in the treatment of HER2-positive gastric cancer by promoting tumor cell mitosis [38]. Genes in the DNA replication initiation pathway including *ORC6* were shown to have prognostic values for numerous cancers including breast cancer [39,40,41,42,43]. Over-production of initiation proteins was widely observed during early stages of tumorigenesis in multiple cancers. ORC6 overexpression has been shown to promote cell proliferation, migration, and invasion of cancer cells [44]. Moreover, down-regulation of initiation of replication genes sensitizes tumor cells to anti-cancer treatment [45]. Therefore, we suggest that *SHCBP1* and *ORC6* are prime targets for anti-cancer interventions in breast cancer treatment.

Our study is the first to combine simultaneous mapping of recurrent DSBs and stable structural breakpoints, along with gene expression, in two well-chosen mammary cell lines. Such a comparison allowed us to specifically test and uncover evidence for recurrent DSBs potentially resulting in structural changes in the chromosome and impacting disease-relevant pathways for breast cancer development. The technological advance and experiment design are readily adaptable to large-scale disease models to detect disease-specific DSB formation and genome instability.

## Figures and Tables

**Figure 1 genes-13-01228-f001:**
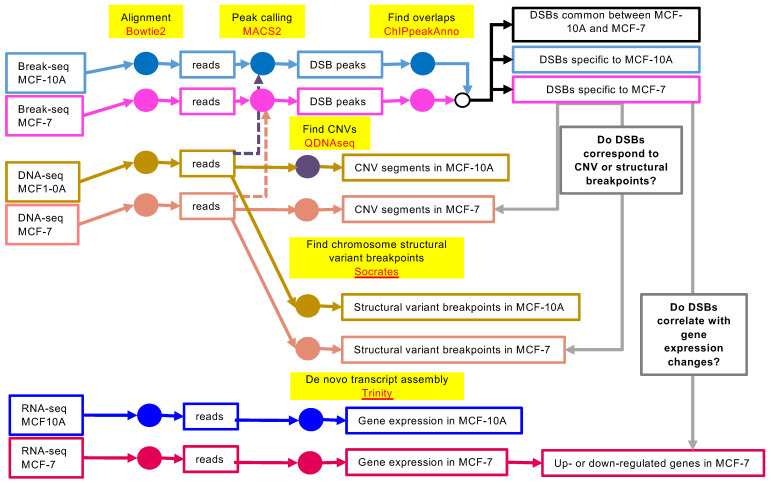
Experimental design. Work flow for multi-dimensional genomic queries of the MCF-7 and MCF-10A cell lines for the identification of cancer-specific chromosome breakage sites and potentially impacted genes. The color-coded nodes denote the analytical steps utilizing the highlighted computational methods.

**Figure 2 genes-13-01228-f002:**
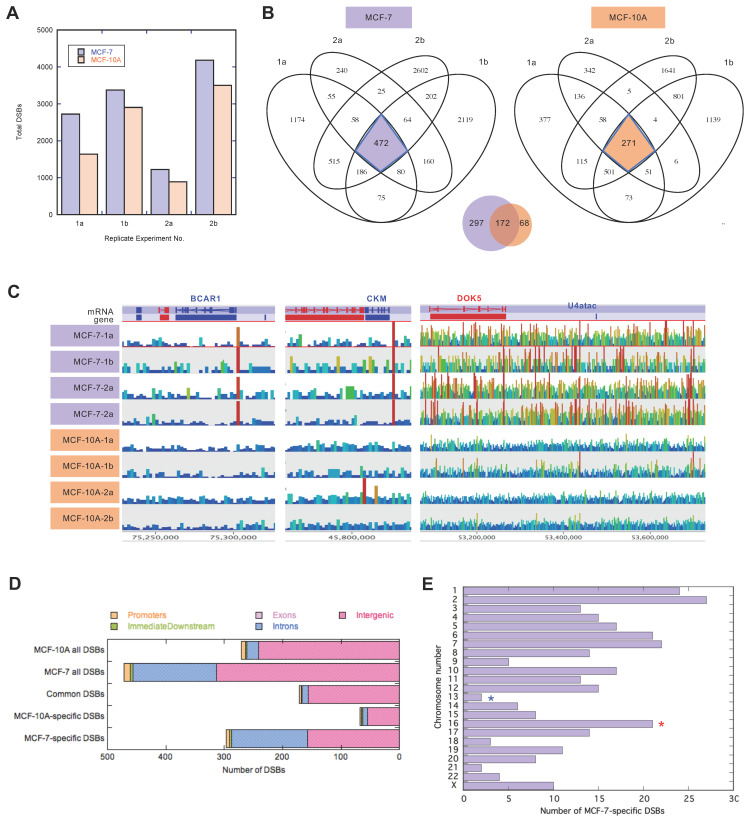
Break-seq analysis identifies cancer-specific chromosome breakage sites in the MCF-7 cell line. (**A**) Number of DSBs identified in each replicate experiment for both MCF-7 and MCF-10A. (**B**) Venn diagrams of consensus DSBs found in all four replicate experiments for MCF-7 and MCF-10A. (**C**) Examples of cancer-specific consensus DSBs in MCF-7 cells and not in MCF-10A cells. The genes proximal to the chromosome breaks are BCAR1, CKM, and DOK5, located on chromosome 16q, 19q, and 20q, respectively. (**D**) Distribution of DSBs overlapping genomic features in each of the five categories as indicated. (**E**) Distribution of DSBs per chromosome. Those chromosomes with the highest and lowest number of DSBs per Mb of DNA are marked by red and blue asterisks, respectively.

**Figure 3 genes-13-01228-f003:**
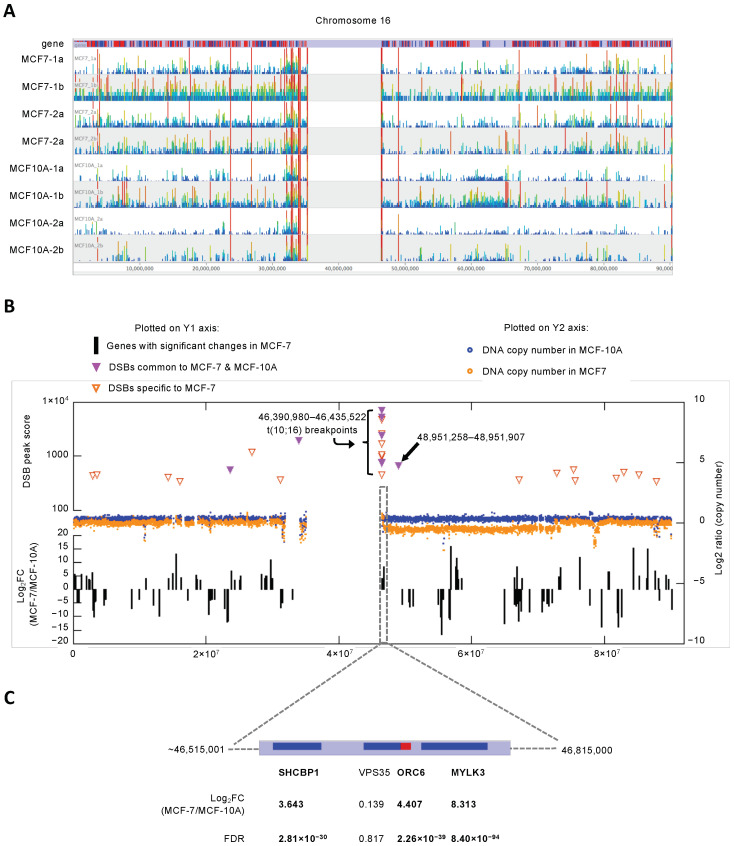
Structural variation and gene expression at the 16q pericentromere. (**A**) Break-seq profiles of all four replicate experiments in MCF-7 and MCF-10A cells on chr16. (**B**) Overlaid plots for DSB scores (top plot), DNA copy number (middle plot), and gene expression (bottom plot) for chr16. The DSB score and gene expression levels expressed as Log2 fold change (FC) in transcript level in MCF-7 over that in MCF-10A cells are plotted on the left, Y1, axis. The DNA copy numbers are plotted on the right, Y2, axis. (**C**) Expanded view of gene cluster immediately downstream of the pericentromeric region of 16q. FDR, false discovery rate.

**Figure 4 genes-13-01228-f004:**
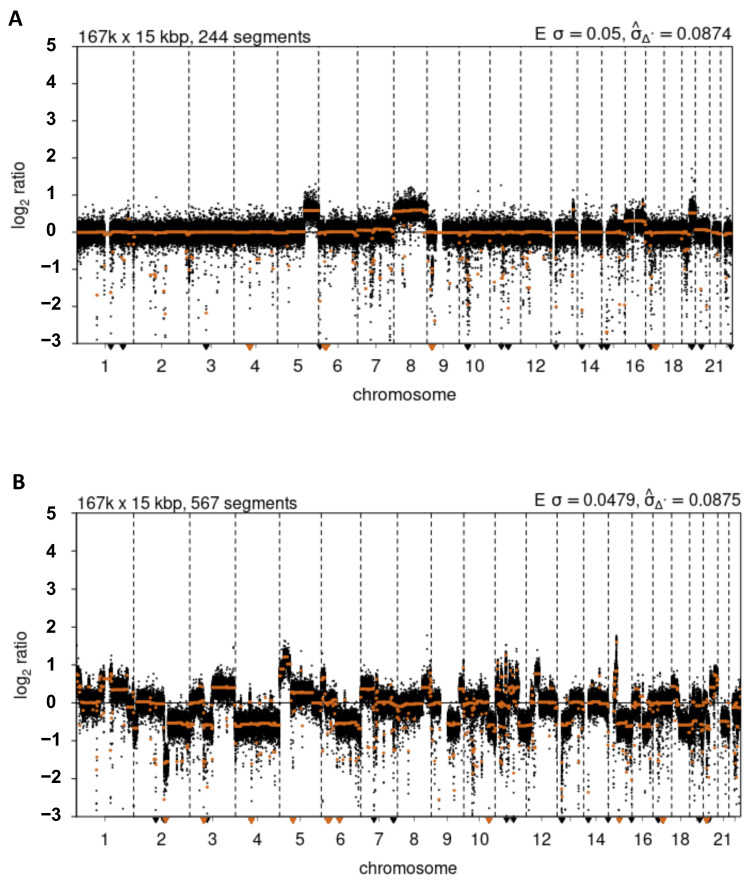
Copy number variations. Copy number profiles for (**A**) MCF-10A and (**B**) MCF-7 cells. Copy number is expressed as Log_2_ transformed normalized sequence read counts in 15 kilobasepair (kbp) segments across the autosomes. Copy number profiles were generated after correction for GC content and mappability, followed by segmenting using default parameters in QDNAseq.

**Figure 5 genes-13-01228-f005:**
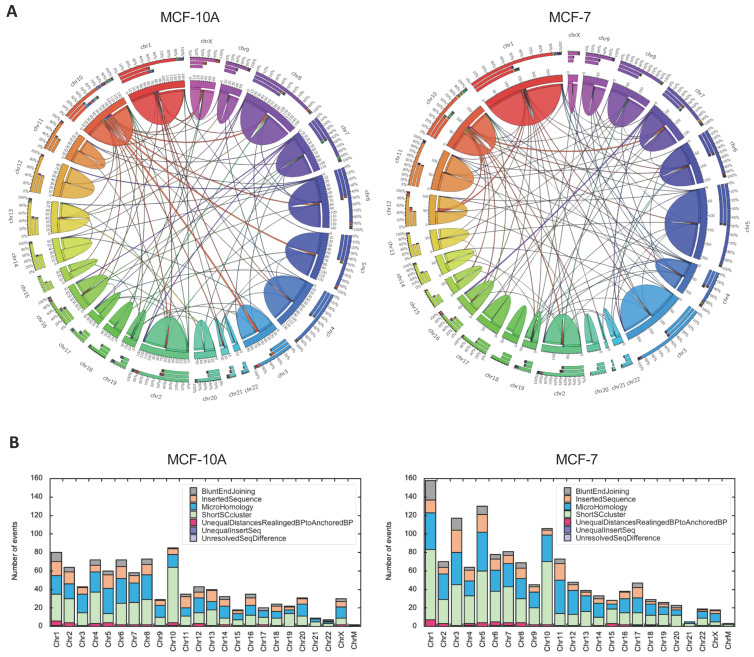
Structural variations in MCF-10A and MCF-7 cell lines. (**A**) Circos displays of paired structural variation events detected by Socrates for MCF-10A (971 events) and MCF-7 (1334 events). Each chromosome is color-coded. Intra-chromosomal breakpoints are represented by the dome above the chromosome; the width of the dome corresponds to the number of events. Inter-chromosomal translocations are represented by ribbons connecting the two translocated chromosomes, with the thickness of the ribbon corresponding to the number of events. The bar graphs beneath the chromosome indicate the relative proportion of intra- (same color of the chromosome) and inter- (color of the connecting chromosome) chromosomal events. (**B**) Structural variants from paired chromosomal translocations were further classified into seven categories as indicated, and plotted as stacked column plots for each chromosome.

**Figure 6 genes-13-01228-f006:**
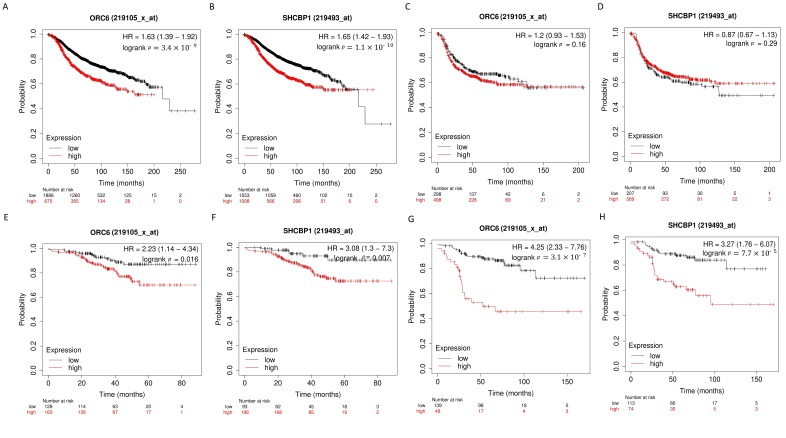
Survival prediction analysis of breast cancer patients. The survival function of progression-free survival (PFS) time is analyzed. (**A**) Kaplan–Meier (K–M) plots for patients with primary ER-positive breast cancer discriminated by *ORC6* expression level into low- and high-risk groups (expression value cut-off = 548). (**B**) K–M plots for patients with primary ER-positive breast cancer discriminated by *SHCBP1* expression level into low- and high-risk groups (expression value cut-off = 157). (**C**) K–M plots for patients with primary ER-negative breast cancer discriminated by *ORC6* expression level into low- and high-risk groups. (**D**) K–M plots for patients with primary ER-negative breast cancer discriminated by *SHCBP1* expression level into low- and high-risk groups. (**E**) K–M plots for patients with primary ER-positive breast cancer discriminated by *ORC6* expression level into low- and high-risk groups (expression value cut-off = 373). Cohort treatment: endocrine therapy + neoadjuvant therapy. (**F**) K–M plots for patients with primary ER-positive breast cancer discriminated by *SHCBP1* expression level into low- and high-risk groups (expression value cut-off = 106). Cohort treatment: endocrine therapy + neoadjuvant therapy. (**G**) K–M plots for patients with primary ER-positive breast cancer discriminated by *ORC6* expression level into low- and high-risk groups. Cohort treatment: endocrine therapy + adjuvant therapy (expression value cut-off = 692). (**H**) K–M plots for patients with primary ER-positive breast cancer discriminated by *ORC6* expression level into low- and high-risk groups (expression value cut-off = 205). Cohort treatment: endocrine therapy + adjuvant therapy. Higher risk (red color line) is associated with higher expression values.

## Data Availability

All data from this study have been deposited in the GEO database under the accession number GSE207716.

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
