# Peer review of "Identification of Recurrent Chromosome Breaks Underlying Structural Rearrangements in Mammary Cancer Cell Lines"

_genes, 2022, doi:10.3390/genes13071228_

Round 1

Reviewer 1 Report

  1. The English need improvement since there are some grammatical and syntax errors in the manuscript. For example,
  • in line number 10, the words “accumulation” may be as “the accumulation”;
  • in line number 49, “essential” as “are essential”;
  • in line number 97, “RNA” as “the RNA”;
  • in line number 107, “false” as “the false”;
  • in line number 110, “genes is” as “genes are”;
  • in line number 112, “100 μl” as “a 100 μl”;
  • in line number 122, “manufacturer's” as “the manufacturer's”;
  • in line number 133, “K-M” as “the K-M”;
  • in line number 145, “pipeline are” as “pipeline is”;
  • in line number 156, “greater” as “a greater”;
  • in line number 196, “structural” as “the structural”;
  • in line number 342, “treatment” as “the treatment”.

The grammar mistakes which are not mentioned here are also to be checked and corrected properly.

  1. There are some typing mistakes as well, and authors are advised to carefully proofread the text. For example,
  • in line number 14, 32 and 142, the words “double strand” may be as “double-strand”;
  • in line number 127, “proposes” as “propose”.

The typos not mentioned here are also to be checked and corrected properly.

  1. Check the abbreviations throughout the manuscript and introduce the abbreviation when the full word appears the first time in the text and then use only the abbreviation (For example, ER-positive, fold change (FC), etc.). And it should be in both abstract as well as in the remaining part of the manuscript. Make a word abbreviated in the article that is repeated at least three times in the text, not all words need to be abbreviated.

  1. The figure legends should be improved and a proper footnote should be given. All legends should have enough description for a reader to understand the figure without having to refer back o the main text of the manuscript. For example, the necessary expansion may be given instead of abbreviations.

  1. The references are not arranged properly in a uniform format and it should be carefully checked and corrected as per the journal instructions. For example, the reference is not properly cited with page numbers (Reference number 12).

Reviewer 2 Report

Review

genes-1733307

“Identification of recurrent chromosome breaks underlying structural rearrangements in mammary cancer cell lines”

Senter et al.

Dear Authors,

The following article utilizes state-of-the-art sequencing methods implicated in the characterization of the correlation between accidental formation of recurrent DNA double-strand breaks and the generation of structural chromosomal rearrangements in MCF7 breast cancer cell line, in contrast to the epithelial non-cancerous cell line - MCF10A. With the help of a highly sophisticated sequencing approach, authors have identified that the generation of concurrent DSBs and structural variation breakpoints occurred exclusively in the pericentromeric region of chromosome 16q in MCF-7 cells. Moreover, authors have additionally characterized a copy number variations at 16q. In summary, authors have proposed a model for DSB-driven chromosome rearrangements that led to translocation of 16q to 10q, and an eventual loss of 16q that does not involve its pericentromeric region.

In addition, based on RNA-seq data, evidences have been accumulated, suggesting that the selected genes (SHCBP1, ORC6 and MYLK3), located in close proximity to the 16q pericentromeric region are up-regulated in MCF7 cells and could display characteristics of potential oncogenes.

Although, the article provides clear evidences supporting the central questions authors are trying to address, there are additional point, which need to be clarified.

Please, find below my concerns and recommendations:

1.      The Introduction section of the manuscript need to provide more clear and comprehensive background of the current topic, as in its present form it is difficult to understand and justify the approach selected by the authors to demonstrate their hypothesis.

2.      The Materials and Methods section should be additionally elaborated and should provide more comprehensive information about the experimental procedures, software and statistical analysis. Moreover, it is not clear how the authors have grown the cells in order to achieve the indicated cell numbers (>4,7x107) and why it was necessary to generate such high number of cells, if the Break-seq and RNA-seq required only 5x106 cells. There is also no information in which cell cycle phase, the cells have been collected for sequencing analysis.

3.       A major concern is that authors have used a MCF7 cell lines, which is characterized with extremely altered karyotype (both numerical and structural chromosomal abnormalities are detectable in high frequency). Despite the fact that there is enormous amount of data characterizing the MCF7 karyotype by mFISH or SKY painting, there is no consensus on the unified karyotype version, which automatically require that the authors should provide detailed karyotype data of the particular MCF7 clone utilized in their study. This is particularly important for the status of the chromosome 16, as multiple reports show enormous multiplication of this chromosome and its involvement in multiple translocations, while other studies report subtle numerical abnormalities or even chromosome loss.

4.      There is a noticeable difference in the number of experimentally identified DSBs between the experimental replicates for MCF7 and MCF10A cells, shown in Figure 2. Could the authors explain the origin of this difference and how this affects the quality of the overall experimental data?

5.      In Figure 4 the authors have shown a detailed map of the copy number variation in all chromosomes. Most of the chromosomes does not show differences between MCF7 and MCF10A cell lines. Could the authors explain how this fit with the diametrically different chromosome number between the investigated cell lines?

6.      The circular plots shown in Figure 5A are beautiful, but due to the considerable amount of information they are carrying, they are very difficult to follow. Moreover, the comparison between MCF10A and MCF7 cells revealed almost no difference in the number of mapped translocations. Could authors clarify this?

7.      Authors have indicated that the whole genome paired-end sequencing has a coverage of >1.8x. What does this really mean; Is the coverage 2x or 20x? However, could the authors discuss and provide additional evidences that the achieved coverage was enough to support the obtained results.

8.      The RNA-seq data revealed an over expression of a certain genes. It will be necessary the up-regulation and the over expression of the selected genes to be confirmed by alternative methods. Authors should also elaborate more on explaining the consequences of such up-regulation.

9.      The above is hold true for the breaks and translocations found on chromosome 16q. Especially the translocation between 16q and 7q or 10q, should be confirmed by alternative methods.

Round 2

Reviewer 2 Report

Review, revision

genes-1733307

“Identification of recurrent chromosome breaks underlying structural rearrangements in mammary cancer cell lines”

Senter et al.

Dear Editor, Dear Authors,

In the revised version of the following article “Identification of recurrent chromosome breaks underlying structural rearrangements in mammary cancer cell lines” by Senter et al. submitted for publication in the genes journal, authors have significantly improved the manuscript and have added additional data to support their conclusions. I am convinced that in its present format the article brings significant information in an easy to follow form, which will be beneficial for the field. Therefore, I would recommend the article for publication in the journal.